# Enhanced Solubility and Stability of Aripiprazole in Binary and Ternary Inclusion Complexes Using Hydroxy Propyl Beta Cyclodextrin (HPβCD) and L-Arginine

**DOI:** 10.3390/molecules28093860

**Published:** 2023-05-03

**Authors:** Sophia Awais, Nouman Farooq, Sharmeen Ata Muhammad, Hamed A. El-Serehy, Farrah Ishtiaq, Mehwish Afridi, Hina Ahsan, Amin Ullah, Tariq Nadeem, Kishwar Sultana

**Affiliations:** 1Department of Pharmacy, Faculty of Pharmacy, University of Lahore, Lahore 54590, Pakistan; 2Department of Pharmaceutical Chemistry, Faculty of Pharmacy, IBADAT International University, Islamabad 44000, Pakistan; 3Department of Medicine, Nishtar Medical University, Multan 66000, Pakistan; 4Department of Zoology, College of Science, King Saud University, Riyadh 11451, Saudi Arabia; 5Cardiac Renal Institute (CaRe Institute), Chubbuck, ID 83202, USA; 6Riphah Institute of Pharmaceutical Sciences, Riphah International University, Islamabad 46000, Pakistan; 7Department of Health and Biological Science, Abasyn University Peshawar, Peshawar 25000, Pakistan; 8Institute of Pathology Lab, University of Cologne, 50923 Koln, Germany; 9National Center of Excellence in Molecular Biology, University of The Punjab, Lahore 54000, Pakistan; 10Department of Pharmacy, Iqra University, Islamabad 75500, Pakistan

**Keywords:** aripiprazole, L-arginine, hydroxypropyl-β-cyclodextrin, scanning electron microscopy, antipsychotic, schizophrenia

## Abstract

The low water solubility of an active pharmaceutical ingredient (aripiprazole) is one of the most critical challenges in pharmaceutical research and development. This antipsychotic drug has an inadequate therapeutic impact because of its minimal and idiosyncratic oral bioavailability to treat schizophrenia. The main objective of this study was to improve the solubility and stability of the antipsychotic drug aripiprazole (ARP) via forming binary as well as ternary inclusion complexes with hydroxypropyl-β-cyclodextrin (HPβCD) and L-Arginine (LA) as solubility enhancers. Physical mixing and lyophilization were used in different molar ratios. The developed formulations were analyzed by saturation solubility analysis, and dissolution studies were performed using the pedal method. The formulations were characterized by FTIR, XRD, DSC, SEM, and TGA. The results showcased that the addition of HPβCD and LA inclusion complexes enhanced the stability, in contrast to the binary formulations and ternary formulations prepared by physical mixing and solvent evaporation. Ternary formulation HLY47 improved dissolution rates by six times in simulated gastric fluid (SGF). However, the effect of LA on the solubility enhancement was concentration-dependent and showed optimal enhancement at the ratio of 1:1:0.27. FTIR spectra showed the bond shifting, which confirmed the formation of new complexes. The surface morphology of complexes in SEM studies showed the rough surface of lyophilization and solvent evaporation products, while physical mixing revealed a comparatively crystalline surface. The exothermic peaks in DSC diffractograms showed diminished peaks previously observed in the diffractogram of pure drug and LA. Lyophilized ternary complexes displayed significantly enhanced thermal stability, as observed from the thermograms of TGA. In conclusion, it was observed that the preparation method and a specific drug-to-polymer and amino acid ratio are critical for achieving high drug solubility and stability. These complexes seem to be promising candidates for novel drug delivery systems development.

## 1. Introduction

Many due to poor solubility and dissolution rates exhibit low bioavailability due to poor solubility and dissolution rates. Hence, solubility enhancement by the use of polymers is the most frequently adopted technique for the development of the drug. A novel technique for the development of solubility involves multicomponent amino acid systems along with cyclodextrin [1]. Aripiprazole is abbreviated as ARP, whose chemical formula can also be written as 7-[4-[4-(2, 3-dichlorophenyl) piperazin-1-yl] butoxy]-3, 4-dihydro-1-H-quinolin-2-one]. It is an atypical antipsychotic medication that has a distinctive mode of action utilized to treat schizophrenia as well as the mania that occurs in type I bipolar disorder [2,3]. It is a quinolinone derivative as well as a BCS Class II candidate. It shows high lipophilicity, such as Log P of 4.55, and a poor aqueous solubility of 10 ± 981.39 ng mL^−1^ [2,4,5]. ARP experiences high first-pass effects upon oral administration, which results in poor therapeutic availability [6,7].

Cyclodextrins, which are abbreviated as CDs, are cyclic glucose oligomers. The hydrophobic internal cavity of CD molecules allows them to develop a non-covalent host–guest complex alongside multiple drug molecules of the appropriate size and shape. The physicochemical properties, such as melting point, aqueous solubility, chemical and physical stability, unpleasant odor or taste, volatility, and drug delivered via biological membranes, can be significantly altered by its inclusion; the encapsulation of a drug in the CD cavity has attracted pharmaceutical interest.

Additionally, CDs are utilized to relieve gastrointestinal irritation as well as to prevent incompatibilities amongst excipients and drug substances. As a result, the CD complex formation improves the dissolution, bioavailability, biological activity, and solubility of the drugs while also significantly improving their pharmaceutical formulations. The three forms of cyclodextrins with 6 (α), 7 (β), or 8 (γ) anhydroglucose units in the ring structure—alpha, beta, as well as gamma—are the most often used. The solubility of ARP could be enhanced by different mechanisms such as inclusion complexes with polymers or the formation of nano complexes. The nano complexes produce ≥75% dissolution rate compared to the samples prepared by physical mixing and co-precipitation at higher ratios, while in case if solid lipid nano particles, the solubility is enhance 1.6 folds [8]. The dissolution enhancement of ARP in the case of nanoparticles, solvent evaporation, and physical mixing, is polymer dose dependent. When the polymer is increased, the solubility is enhanced, and vice versa [9,10].

Amino acids comprise many properties which are adequate for pharmaceutical applications, and they interact adequately with CDs. They form hydrogen bonding and salt formation or electrostatic interactions with the CDs and drug, respectively, and are used to overcome the non-desirable qualities of the drug. An increase in antimicrobial activity and reduced toxicity are a couple of examples found in the literature [11,12].

Arginine is frequently employed because it has prompt release properties and can enhance stability, solubility, bioavailability, and absorption. Complexation of L-arginine (LA) and randomly-M-β-CD with oxaprozin improve drug stability and solubility [13].

This study sought to obtain and thoroughly characterize the inclusion complex of Aripiprazole with HPβ-CD to improve the ARP solubility. The binary system prepared by solvent evaporation and lyophilization showed better solubility results than the physical mixture. However, the introduction of a ternary compound, L-Arginine, exhibits a remarkable increase in solubility; hence, maintaining the advantage of lyophilization. This is a novel technique for the solubility enhancement of ARP. Binary complexes prepared by different methods have previously been investigated but the ternary inclusion complex with ARP-HPβCD is used here for the first time. Furthermore, the exact ratio for the maximum solubility is also calculated, which is far less than the higher molar ratios prepared by physical mixing. The solvent evaporation method produced comparable results but the lyophilization method exhibited a remarkable response. The usual trend of increased solubility by increasing the amount of polymer is not followed in lyophilized inclusion complexes. The molar ratio 1:1:0.27 gave much higher results than the 1:9:1 molar ratio—82 times more than solvent evaporation and physical mixing. This is even higher than the nano milled powder of aripiprazole. By doing so, the amounts of the drug required, cyclodextrin, and the amino acid utilized are minimized, leading to fewer side effects and increased cost effectiveness. Although the lyophilization method is superior, solvent evaporation is much cheaper and comparatively better than physical mixing. Binary and ternary ratios were selected on the basis of quality by design (QbD). 

The complexes were studied using AT-FTIR, scanning electron microscopy, X-ray powder diffractometry, differential scanning calorimetry, and thermogravimetric analyses. The physicochemical performance of the obtained products was further investigated using the equilibrium solubility study and dissolution studies.

## 2. Results

### 2.1. Solubility Study

#### 2.1.1. Physical Mixing

In Table 1, the samples HPM-32, HPM-33, HPM-34, and HPM-35 gave dissolution values of 1.72, 24.99, 22.32, and 19.81 (µg/mL), respectively. While samples HPM-36, HPM-37, HPM-38, HPM-39, HPM-40, and HPM-41 gave increased solubility values of 9.88, 1.34, 1.01, 23.33, 9.8, and 1.067 (µg/mL), respectively (Figure 1). 

#### 2.1.2. Lyophilization

This method showed different behavior in solubility after complexation as shown in Table 1. Binary complexes 42, 44, and 45 gave solubility values of approximately 0.12, 1.02, and 0.78 (in µg/mL), respectively, while in the case of ternary complexes, samples HLY-46, HLY-47, HLY-48, HLY-49, HLY-50, and HLY-51 showed 2.04, 20.45, 12.6, 10.20, 5.60, and 7.73 (in µg/mL), respectively. 

#### 2.1.3. Solvent Evaporation

Binary complexes HSE-52, HSE-54, and HSE-55 gave 18.38, 0.9, and 1.89 times (µg/mL) improved solubility, respectively. Samples HSE-56, HSE-57, HSE-58, HSE-59, HSE-60, and HSE-HSE-61 gave 20.91, 1.63, 1.42, 24.9, 15.1, and 1.63 times enhanced solubility (Table 1).

### 2.2. Dissolution Studies

HPM32, HPM33, HPM34, and HPM35 gave the dissolution values 0.61, 0.45, 0.41, and 0.36 (in µg/mL). At the same time, the ternary complexes HPM36, HPM37, HPM38, HPM39, HPM40, and HPM41 give 0.48, 0.025, 0.57, 0.55, 1.2, and 0.49 (in µg/mL). HLY 42 showed a value of 0.4 (in µg/mL), while ternary complexes HLY47, HLY48, and HLY51 give the dissolution values of 0.62, 0.03, and 0.05 (in µg/mL). In solvent evaporation, HSC52, HSC54 and HSC55 gives dissolution values of 1.29, 1.42 and 0.66 (in µg/mL). In contrast, the ternary complexes HSC56, HSC57, HSC58, HSC59, and HSC60 give dissolution values of 0.01, 1.25, 1.05, 0.52, and 1.38 (in µg/mL) as illustrated in Figure 2.

### 2.3. Attenuated Total Reflection Fourier Transform Infrared Spectroscopy (ATR-FTIR)

FT-IR spectroscopy is used to identify any change, widening, or disappearance in the intensity of the peaks and bond shifting, indicating the formation of a new compound due to a reduction in the stretching vibrations of the ARP molecule due to its accumulation in the CD cavity. In Figure 2A, ATR-FTIR spectra of the ARP gave characteristic peaks at 1672 cm^−1^ (carbonyl peak), 1027 cm^−1^ (for C-O-C stretch), as well as 763 cm^−1^ (for C-Cl stretching). HPβCD spectra gave peaks at 1080.12 (C-O-C stretch), 2910 cm^−1^ (for O-H stretching), 2910 cm^−1^ (CH, stretch), and 1362.68 cm-1 (for OH bending vibrations). LA showed a peak at 3251 cm^−1^ (for broad OH carboxylic), 3067 cm^−1^ (for NH guanidine), 2860 cm^−1^ (CH stretch), 1318 cm^−1^ (CN stretch), and 1563 cm^−1^ (presence of amide).

Figure 3A shows the FT-IR spectrum of the physical mixture HPM 39 and 41. It is observed that the spectra are a combination of HP-β-CD, LA, and ARP spectra. This observation showed that there is a slight change in the bond angles. The unique pattern in the FT-IR spectra in Figure 3B–D of HLY samples (HLY 42, 45, 48, 49. 50, 51) indicates the constitutions of inclusion complexes. The complexes prepared by the solvent evaporation method (HSE 54, 55, 60, and 61) demonstrate the same results. In Figure 3E,F, no characteristic peak of ARP indicates the complete inclusion of the drug into CD cavity. The peak of LA is also shifted, indicating that the ARP and LA are embedded into the CD cavity. Other characteristic peaks are also diminished or shifted, which indicates the formation of the new compound.

### 2.4. Scanning Electron Microscopy (SEM)

SEM images of ARP, binary, and ternary complexes prepared by physical mixing, lyophilization, and the solvent evaporation method are shown in Figure 4. ARP appeared as flake crystals, HPβCD appeared as spherical particles with an amorphous character, and LA appeared as non-linear crystals, which followed the previously reported results. ARP is crystalline, HPβCD is totally amorphous, LA appears as crystals, HPM35, HSE55, HLY42, HPM40, HSE57, and HLY47 demonstrate a change in drug morphology, while HLY47 demonstrates a total change in drug structure by completely transforming into an amorphous form. 

### 2.5. X-ray Diffractometry (XRD)

XRD patterns of aripiprazole, HPβCD, L-Arginine, and different formulations are given in Figure 5 for comparative study. ARP is crystalline with 8.65, 11.55, 16.25, and 17.65, with higher intensity peaks at 20.25, 20.3, 20.35, 23.35, and 24.90. LA demonstrated the nature of a crystal structure with 2θ distinct peaks at 11.2°, 14.7°, 16.45°, 17.25°, 18.45°, and 19.25°, with the higher peak intensity at 23.05° and 27.55°. However, HPβCD is amorphous. Graph A shows a decrease in the peak intensity of the drug, and graph B in Figure 6 demonstrates the change of the drug to an amorphous form. However, graph C shows a considerable decrease in peak intensity, and graph D shows the amorphous nature of the complex. 

### 2.6. Thermogravimetric Analysis (TGA)

The thermograms obtained gave us the idea of thermal stability. The sample was heated from 30.00 °C to 500.00 °C at 10 °C per min under a dry nitrogen gas with a flow rate of 20 mL per min using PerkinElmer STA 6000 simultaneous TGA/DSC analyzer. Sample HPM35, HLY45, HPM39, and HLY47 were evaluated using TGA thermograms as shown in Figure 7.

### 2.7. Differential Scanning Calorimetry (DSC)

The powder DSC patterns were conducted to rule out the amorphous and crystalline structure of the binary and ternary samples (ARP, HPβCD, and LA; ARP, HPβCD, and LA), while the ratio of amorphous–crystalline was also studied. Physical mixtures show a decrease in amorphous nature as highlighted in Figure 8. Solvent evaporation shows better results than physical mixing, while in the case of lyophilized samples, especially HLY47, the drug is completely transformed into an amorphous form as depicted in Figure 9.

## 3. Discussion

Aripiprazole (ARP), a weak basic drug, has been found to have pH-dependent solubility. According to preliminary investigations, ARP was more drug-soluble in greater acidic environments. A pH solution over 6.8 showed a very low solubility of ARP, while ARP had high solubility when the pH values were 1.2 and 4.0. To increase the dissolution and solubility of ARP, suitable hydrophilic polymers and amino acids were screened. To assess the solubility of the drug in each solution, the different amino acids and polymers were made as 1% aqueous solutions. To assess ARP’s solubility in each amino acid and polymer, UV spectroscopy was used. The samples were studied at a wavelength of 249 nm, which Silki and Sinha employed in 2018. Except for a few, all the amino acids significantly improved the drug solubility as compared with the pure drug. ARP solubility with polymers was indeed dependent on the polymer type. HPβCD, the polymer chosen for this study, seemed to have the highest solubility and was thus selected for solid dispersions [14].

Following a 72-h equilibrium solubility Study with HPβCD Complexes, equilibrium solubility of ARP in 1, 5, and 10% (m/V) LA solutions were conducted. The statistical study showed that the ARP concentrations between 1 and 5 percent (m/V) LA solutions differed significantly, but the ARP concentrations varied significantly across all LA solutions. According to the linear regression analysis, the concentration of ARP increased approximately linearly as the concentration of LA increased (R2 = 0.9522, for example) [15]. We also investigated how the concentration of LA affects the solubility of ARP. LA and ARP concentrations were found to be linearly correlated in their investigation, even though the study of LA concentrations was lower (0.001–0.1% (m/V)) [16]. It has been demonstrated that the free amino acid dissolves more slowly than the ARP-HPBCD-LA. Furthermore, the inclusion complexes at acidic pH appeared to retain a higher total amount dissolved than the free acid.

The pharmaceutical sector uses solvent evaporation to create solid dispersions (SDs) [14,17]. Solid dispersions are formed using the method by which the drug is dispersed within the hydrophilic polymeric carrier as the components dissolve and combine along the mixing length. The solid dispersions promote drug release in the aqueous media, further enhancing bioavailability. Aripiprazole displays the effect of the different formulations, both with and without an amino acid. Compared with the pure drug, all formulations displayed higher levels of drug release and solubility. Moreover, changes in the drug release profiles and solubility were explained by polymer and acidifier content variations. When compared with the pure drug and perhaps other formulations, the acid booster-containing formulations, such as ARP: HPβCD:LA (1:3.6:3.6), ARP: HPβCD:LA (1:1:0.27), ARP: HPβCD:LA (1:9:1), and ARP: HPβCD:LA (1:3.6:3.6), showed the maximum levels of drug solubility and indeed better profiles of drug release such as >90%. These formulations also decreased the dissolution media pH from 4.5 to 5.5, instead of 6.75 for the pure drug. Because of the boost in acidity, ARP: HPβCD: LA (1:3.6:3.6) was 3.6-part amino acid and demonstrated somewhat more significant drug release than ARP: HPβCD (1:1). Compared with the pure drug, ARP: HPβCD: LA exhibited greater solubility, such as 13.8 μg/mL, along with drug release, which was 13–37% (Figure 1)

Drug release, however, has been revealed to be influenced by the drug to the carrier ratio. Hence, the more carriers there are, the more a drug transforms into its amorphous form, increasing the amount of drug released. These similar outcomes were seen in the analysis with binary combinations that were amino acid free. Figure 1 revealed a clear separation between the HPM1 through HPM5 and HLY 42- HLY46 release profiles, but HPM1 through HPM5 release profiles were comparable. The ratio of the drug to the polymer was the apparent cause of the variation for HPM1 through HPM5. A total of 90% HPβCD and 10% aripiprazole were present in HSE 52 and HSE 61. The content for HSE-52 was made up of 50% HPβCD and 50% ARP. Fewer levels of polymer were present, which could have caused the greater drug content of the solid dispersions to become saturated. HSE57 and HSE58 exhibited higher rates of drug release (Figure 2) and were unsaturated solid dispersions with even more hydrophilic HPβCD [18,19,20,21,22].

AT-FTIR spectra (Figure 3) were obtained in Figure 2 for pure components and variously prepared binary and ternary combinations in the 1500 to 1800 cm^−1^ spectral range. Despite being somewhat masked by the HPβCD band, the quartet of ARP carbonyl bands stretched among 1690 cm^−1^, and 1594 cm^−1^ still was discernible but did not exhibit any change in binary as well as ternary physical combinations. On the other hand, as a likely outcome of salt formation, binary HPβCD-ARP- inclusion complexes showed marked spectral alterations with regard to the physical combination. While physical inclusion complexes for ARP-HPβCD systems appeared similar to the physical combination, a difference was seen in the solvent evaporation spectrum, where the band disappeared at 1728 cm^−1^. Other authors have noted similar spectral alterations in ARP-βCD lyophilized systems, which they have attributed to the inclusion of complexation-induced dissociation of hydrogen bonds in ARP. Due to the strong interactions among the components, a single, irregular wider band was seen for the ternary solvent evaporation system. The spectra of the systems, on the other hand, which were achieved by introducing the third component into the binary evaporation inclusion complexes, were simply a superposition of their constituents [23]. When the same molar ratios were prepared by lyophilization method, the bond length and the peak intensity were also changed. This is probably due to the complete inclusion of guest molecules into the cyclodextrin cavity. It can be interpreted from these data that the lyophilization method is the superior method for the formation of inclusion complexes [24,25].

The SEM results in Figure 4 represent that the crystalline nature of the aripiprazole is completely transformed to amorphous. The HPβCD is porous, spherical in shape, and porous upon the formation of complexes; its structure changes to a regular folded shape, having the drug and amino acid completely encapsulated in case of lyophilization method. The ternary complexes demonstrate superior results to the binary complexes. It is also observed that the lyophilized mixture with the ratio of 1:1:0.27 (Figure 5) was again better than the other compounds [25,26,27].

X-ray powder diffraction patterns in Figure 6 are used to measure the crystallinity of the complex. The greater the crystallinity (sharp peaks), the lesser the solubility of the drug complex and vice versa [25]. The XRD patterns showed that they are pure elements, along with several binary and ternary solid complexes. Both LA and ARP spectra revealed several intense diffraction peaks, indicating that they are crystalline, but the HPβCD spectrum displayed the flat pattern typical of the amorphous materials. Crystallinity was lost in the solvent evaporation and lyophilization inclusion complexes, whereas the physical inclusion complexes’ spectra revealed the establishment of a unique crystalline phase. Regarding the ARP-LA binary system, the peaks of the physical combination were just the superposition of the pure elements. The drug crystallinity drastically decreased when it was in a physical mixture with HPβCD, while both evaporation and lyophilization spectra showed nearly completely amorphous patterns [25,28]. Regarding the ternary system, some of the most substantial ARP diffraction impacts could be observed in the physical mixture, arising from the HPβCD amorphous profile and disappearing following evaporation or lyophilization. The lyophilized samples have diminished crystal lattice properties [28]. Eventually, it was revealed that the spectrum of the ternary system was produced by introducing LA into ARP-CD evaporation and lyophilization.

TGA is used for checking the thermal stability of drug complexes. TGA curves in Figure 7, representing pure constituents with their different combinations, were obtained. ARP was found to be in a crystalline, anhydrous state by DSC analysis, and TGA analysis indicated its incipient decomposition occurring at 220 °C [29]. LA DSC profile revealed three separate endotherms with peaks at 98, 220, and 244 °C which were each attributable to loss of water through a small segment of LA.2H_2_O, already present in the sample, which melts along with anhydrous LA decomposition which completely decomposes the melt. It was attributed based on the TGA outcome. HPβCD DSC curve appeared as an amorphous substance with weakly bound H_2_O [25]. TGA was used to investigate further the thermal activity of the ARP-HPβCD, as well as ARP- HPβCD-LA ternary system, which had once been characterized through DSC analysis. For the ARP- HPβCD physical mixture, a loss of mass between 35 and 100 °C from amino acid H_2_O escape and T_onset_ = 208 °C from thermal decomposition were noted [9]. The dehydration impact, however, was not present in the evaporated inclusion complexes or lyophilized inclusion complexes as a result of the treatment of the sample. The TGA curves of evaporated or lyophilized inclusion complexes showed a similar impact. In contrast, no mass losses were noticed despite the temperature at which the medication melts.

Differential scanning calorimetry studies (Figure 8) examined the physical state of each component in the sample. When an ARP melting peak is absent, as in the inclusion complex, a conclusion is made that the drug must be in an amorphous form. Three months later, no traces of crystallization were seen, signifying the product stability [8]. A broad endotherm is indicative of loss of water at room temperature and 150 °C, as shown in thermograms of inclusion complexes. The pure API exists in the crystalline form, as this is the API’s thermodynamically stable form at room temperature, according to the X-ray diffractograms and DSC curves. This suggests that the mixes were homogenous single-phase systems and that intermolecular interactions might have developed between the constituent parts [10]. The co-amorphous blends with LA as the co-former demonstrated much higher Tg values than theoretically anticipated. It has been noted that amorphousness production causes the Tg of the inclusion complexes to have its most significant value at a higher molar ratio (Figure 9). In addition, LA is present at the 1:2 molar ratio as an excess amorphous component, and no other interactions among the components take place. According to the results above, colossal porosity, the inclusion of powder complexes with high surface areas, and the drug amorphous state should be considered to promote drug dissolution.

Current findings show that the intensity decreases, as well as the marked broadening or absence of the ARP fusion endotherm, as observed in the DSC evaluations of the ARP-HPβCD inclusion complexes, respectively, were solely caused by their partial or complete amorphization and complexation inside the HPβCD matrix.

## 4. Materials and Methods

In this study, polymer HPβCD was used for the preparation of binary complexes with ARP by physical mixing, solvent evaporation, and lyophilization method in a specified ratio, selected by quality by design (QbD), in the molar ratio of 1:1, 1:2.5, 1:4, and 1:9. For the ternary complexes, amino acid LA was used along with ARP and polymer HPβCD in the same specified ratios of 1:1:1, 1:1:0.27, 1:2.5:0.27, 1:3.6:3.6, 1:4:1, and 1:9:1 as shown in Table 2.

### 4.1. Preparation of Inclusion Complexes by Physical Mixing

The constituents in a specified ratio were softly mixed in a glass mortar and pestle for 5–7 min until homogenized. This mixture is passed through a sieve of 80 µm. This mixture was transferred to properly washed and dried amber-colored glass bottles and labeled carefully. They are placed in desiccator with silica.

### 4.2. Preparation of Inclusion Complexes by Solvent Evaporation Method

The specified quantities of ARP were dissolved in absolute methanol until a crystal-clear solution was obtained for the preparation. Similarly, the polymer was dissolved in sufficient absolute methanol to obtain a clear solution. The two solutions were mixed and placed on a magnetic stirrer. This solution was transferred to an orbital shaker at 150 rpm for 24 h. The solution was transferred to a round bottom flask, and the excess solvent was evaporated using a rotary evaporator. The sample was recovered from the rotary evaporator flask using a minimum solvent. The recovered sample was placed in a petri dish in an oven at 27 °C until completely dried. The dried sample was removed from the petri dish and gently ground with a pestle and mortar. The ground sample was passed through sieve no. 180 µm, and the sample was finally stored in clean and dried amber-colored glass bottles. These amber bottles were kept in desiccators at a temperature of 25 °C.

### 4.3. Preparation of Inclusion Complexes by Lyophilization Method

The specified quantities of ARP were dissolved in absolute methanol throughout the preparation until a crystal-clear solution was obtained. Similarly, the polymer was dissolved in sufficient absolute methanol to obtain a clear solution. The two solutions were mixed and placed on a magnetic stirrer. This solution was transferred to an orbital shaker at 150 rpm for 24 h. The solution was transferred to a round bottom flask, and the excess solvent was evaporated using a rotary evaporator. The samples were recovered from the rotary evaporator flask using the minimum amount of solvent and placed overnight in a scientific freezer at −60 °C. Lyophilizer was used to freeze dry the samples, and the dried powder was sieved through a diameter of 80 µm. The final product was then transferred to amber-colored glass bottles.

### 4.4. Preparation of Capsules

The precisely measured 15 mg of all samples and pure ARP was packed in a hard gelatin capsule. Each sample was made in triplicate. 

### 4.5. In-Vitro Evaluation and Characterization

#### 4.5.1. Solubility Study

Phase solubility study was performed according to Higuchi and Conners, 1965. Accurately weighed samples (0.4 g) were transferred to a glass test tube containing 10 mL of deionized water. Samples were shaken for 2 min on a vortex mixture at about 1400 rpm. The samples were fixed on a shaking incubator at 37 °C for 3 days at 150 rpm and centrifuged for 30 min at 6000 rpm. The supernatant liquid was decanted in a separate glass test tube. With the help of a syringe filter (0.45 µm), 1 mL of the upper clear layer was diluted to 5 mL with deionized water. These samples were analyzed for ARP content at 219 nm using UV–Vis 1600 (Shimadzu Spectrophotometer, Tokyo, Japan). Each reading is measured in triplicate to reduce the chances of errors. Control testing was performed to authenticate the results. The apparent stability constants (*K_s_*) and complexation efficiency (*C.E*) of the samples were calculated using the following equation:$$Ks=\frac{Slope}{So\;\left(I-Slope\right)}$$
where *S_o_* is the equilibrium solubility of ARP in water. $C.E=\frac{Slope}{\left(I-Slope\right)}$.

#### 4.5.2. In Vitro Dissolution Study

The efficacy of most of the oral dosage forms depends upon their dissolution in gastrointestinal fluids before they can be absorbed systemically. It gives the rate of release of drug from the product [30]. All the samples were enclosed in the capsule shells, and the dissolution test was performed using the pedal method (USP 41 Dissolution Apparatus (II), 2018). The SGF was used as a dissolution medium, and its temperature was maintained at 37.0 ± 5 °C. For the preparation of simulated gastric fluid (SGF), 2 g of sodium chloride and 3.2 g of purified pepsin was added to 7.0 mL of HCl, and sufficient water was added to make 1000 mL (USP). With the help of a pipette, 10 mL of the sample was taken out after 5, 15, 30, 60, and 120 min by a syringe filter (0.45 µm). Fresh 10 mL SGF must be added after each sample was taken. Samples were analyzed at 219 nm using UV1700 (Shimadzu Kyoto, Japan) [30,31]. Percentage dissolution efficiency (*DE*) was determined using the trapezoidal method:$$DE\%=\frac{\int oyXdt}{y100Xt}\times 100\%$$
where *y* is the % of the dissolved drug, the area under the dissolution curve is *DE*. *y*100, between time points *t*1 and *t*2, is expressed as a % of the curve at maximum dissolution over the same period. It was computed to compare the performance of binary and ternary formulations [32]. The drug dissolution profile gave the time taken for 50% drug release (T50%).

### 4.6. Methods of Characterization

#### 4.6.1. Attenuated Total Reflection Fourier Transform Infrared Spectroscopy (ATR-FTIR)

The AT-FTIR gives us the information about the formation of new bonds formed. If the bond angles are not shifted, it means that there were no or weak interactions of drug and amino acid with polymer. However, if the bond shifting occurs, it indicates that the drug is encapsulated into the cyclodextrin cavity due to conformational restriction. This leads to the reduced free movement of encapsulated molecules, ultimately leading to a decrease in peak intensity. If the peak intensity is increased, it shows convolution of peaks. ATR-FTIR tests were performed using a broker spectrophotometer and ATR-FTIR 7600 spectrometer (Lambda, Lambda Scientific Systems, Inc., Miami, FL, USA). The ATR-FTIR spectra of the drug, polymer, amino acid, and samples were obtained in the 500–4000 cm^−1^. Raman spectra of inorganic and coordination compounds was consulted for data interpretation. ATR-FTIR provides comprehensive information about the molecular structure of the materials.

#### 4.6.2. Powder X-ray Diffraction

Powder XRD analysis was performed to rule out the chances of crystalline structure using an X-Rays Diffractometer (JDX-3532 JEOL, Tokyo, Japan). The scanning range was performed between 5 and 50 °C, and the rate was 1°/min.

#### 4.6.3. Thermogravimetric Analysis (TGA)

TGA thermograms enable quantitative detection of all the processes by which energy is produced or required (thermal stability). It gives important information about phase transitions, absorption, adsorption, chemosorption, oxidation, reduction, etc., by calculating any change in its mass if the heat is applied at a constant rate. TGA analysis was performed using a PerkinElmer STA 6000 simultaneous TGA/DSC analyzer. The process was performed by heating at 10 °C per min from 30 °C to 500 °C under a dry nitrogen gas with a flow rate of 20 mL/min [31,33]. 

#### 4.6.4. Differential Scanning Calorimetry (DSC)

The qualitative and quantitative thermal properties of solid materials can be found by DSC. Temperature is plotted on the x-axis and the signal obtained is plotted on the y-axis. DSC thermograms of samples were obtained using a PerkinElmer STA 6000 simultaneous TGA/DSC analyzer. The process was performed by heating at 10 °C per min from 30 °C to 500 °C under a dry nitrogen gas with a flow rate of 20 mL/min [33,34].

#### 4.6.5. Scanning Electron Microscopy (SEM)

Micro or surface morphology analyses of pure ARP and binary and ternary complexes were performed using a scanning electron microscope SEM (Perkin Elmer, Waltham, MA, USA).

## 5. Conclusions

For poorly water-soluble drugs with pH-dependent solubility, solid dispersions using the solvent evaporation process and lyophilization are the two methods that have each successfully enhanced the dissolution behavior of the drug and its bioavailability. 

The unique aspect of this study was the comparison between the binary complexes of ARP and HPβCD prepared by physical mixing, solvent evaporation, and the lyophilization method. Furthermore, the effects of LA on the ternary complexes using ARP and HPβCD, prepared by all the selected methods, are also compared. 

Surprisingly, the addition of amino acid changed the conventional behavior of the inclusion complexes, which usually show a proportional increase in dissolution with the increasing amount of polymer or drug. These days, the samples prepared by nano technology also exhibit the same proportional response. However, the addition of LA in ternary complexes not only enhanced the dissolution to a greater extent compared to binary complexes, both in the case of solvent evaporation and lyophilization; it also reflected that the specific molar ratio of 1:1:0.27 has the maximum solubility. If we increase the polymer or the method applied for preparation, the solubility will decrease. 

This is also evident from the characterization results that ternary complexes prepared by lyophilization are more amorphous, stable, and totally encapsulated in the polymer cavity as compared to complexes prepared by the solvent evaporation method. These ternary compounds—with respect to both solvent evaporation and lyophilization—are better than their corresponding binary compounds, even at a lower molar ratio. The solvent evaporation products are even superior to the samples prepared by physical mixing. In comparison to pure aripiprazole, all formulations showed better solubility as well as drug release of the SDs.

Therefore, it can be concluded that the lyophilization method is superior and HPβCD will increase the solubility of the drug at a lower ratio when LA is used as an auxiliary component in a specified ratio, making HLY47 the most promising, cost effective, and more stable candidate for the enhancing solubility of the ARP, as well as the one with the lesser side effects. Furthermore, the amount of polymer used in achieving higher results of solubility has also decreased when compared to the other techniques employed until now. Thus, LA is an excellent auxiliary compound and lyophilization is the best method. 

## Figures and Tables

**Figure 1 molecules-28-03860-f001:**
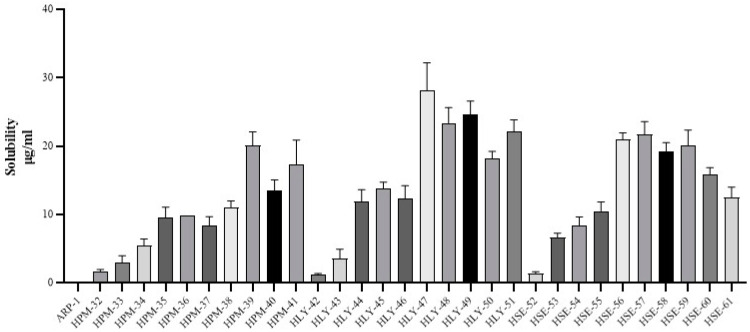
Shows that the solubility of the drug in all the binary and ternary physical mixtures (HPM33, HPM35, HPM37, and HPM39), Lyophilized complexes (HLY43, HLY45, HLY47, HLY49, and HLY51), and complexes prepared by solvent evaporation increases with the increasing amount of cyclodextrin. The addition of LA has remarkably increased the solubility effects, even at a lower ratio of cyclodextrin. HLY47, which is the lyophilized ternary compound with the ratio of 1:1:0.27, is the highest among all samples. At the same time, the sample prepared by solvent evaporation method HSE57 with the same ratio shows a similar behavior. Furthermore, the lyophilized method is superior to the solvent evaporation method.

**Figure 2 molecules-28-03860-f002:**
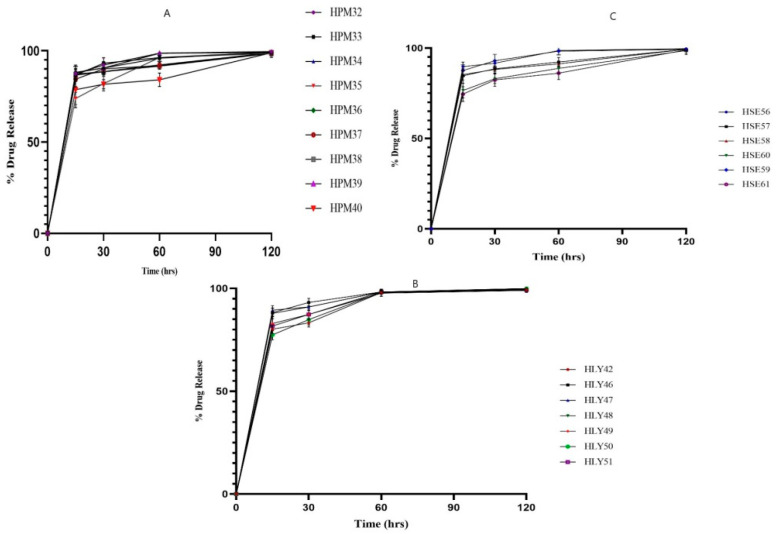
Dissolution studies (**A**) In vitro drug release of HPM 32, HPM-33, HPM-34, HPM-35; ARP dissolution is increased by increasing the ratio of polymer in binary complexes and HPM-41, HPM-40, HPM-39, HPM-38, HPM-37, HPM-3 shows the effect of LA on solubility enhancement. release of (**B**) In vitro drug release of HLY-46, HLY-47, HLY48, HLY-49, HLY-50, and HLY-51, showing a remarkable increase in solubility. (**C**) In vitro drug release of HSE-56, HSE-57, HSE-58, HSE-59, HSE-60, HSE-61 shows the comparative increase in solubility.

**Figure 3 molecules-28-03860-f003:**
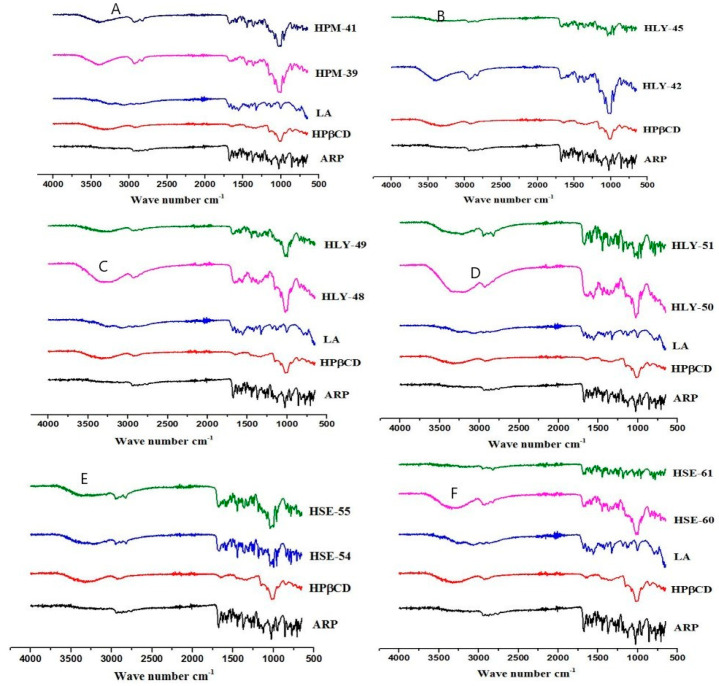
(**A**) Physical mixtures; peak intensity is changed. (**B**–**D**) Complexes by lyophilized complexes reflecting the change in bond length. (**E**,**F**) Complexes by solvent evaporation method exhibiting the formation of a new compound by showing the change in peak intensity.

**Figure 4 molecules-28-03860-f004:**
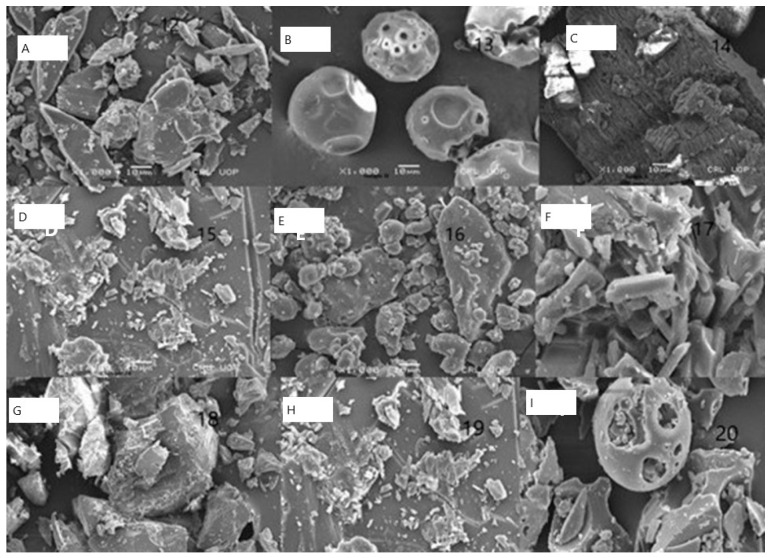
Scanning electron microscopy. (**A**) ARP (**B**) HPβCD (**C**) LA (**D**) HPM35 (**E**) HSE55 (**F**) HLY42 (**G**) HPM40 (**H**) HSE57 (**I**) HLY47.

**Figure 5 molecules-28-03860-f005:**
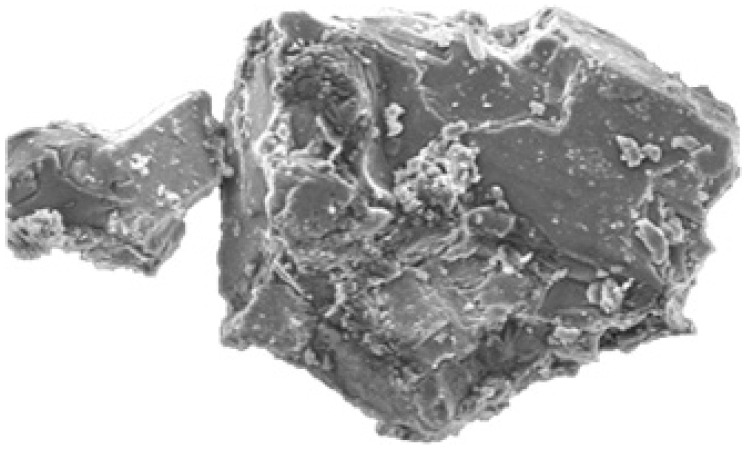
The lyophilization method resulted in a complete transition of drug morphology with an amorphous character.

**Figure 6 molecules-28-03860-f006:**
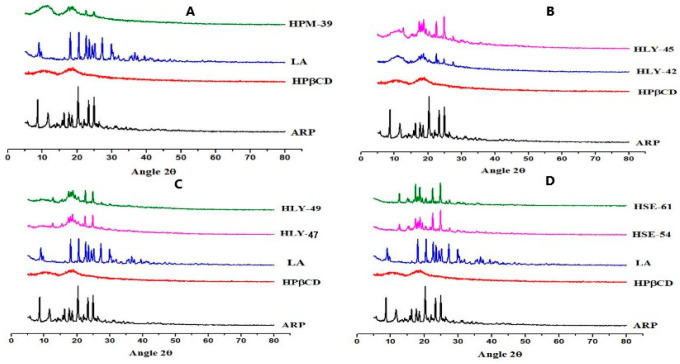
Comparison of XRD graph with complexes: (**A**) XRD of the physical mixture; (**B**) XRD of lyophilized binary complexes; (**C**) XRD of lyophilized ternary complexes; and (**D**) XRD of complexes solvent evaporation method.

**Figure 7 molecules-28-03860-f007:**
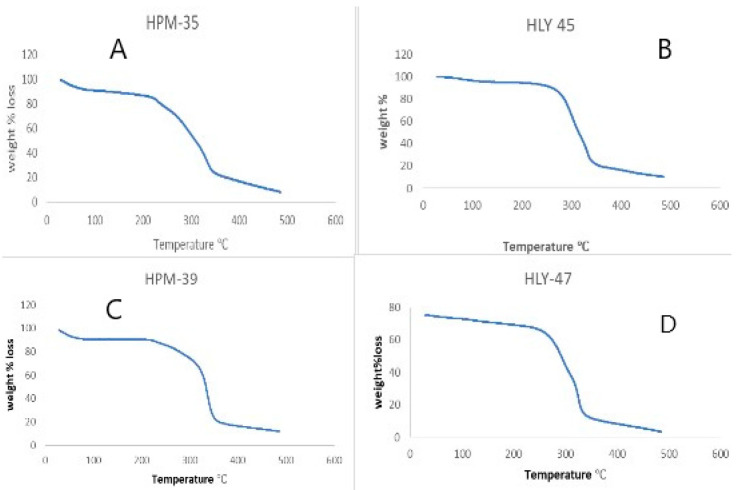
Comparison of TGA graphs: (**A**) HPM35; (**B**) HLY45; (**C**) HPM39; (**D**) HLY37.

**Figure 8 molecules-28-03860-f008:**
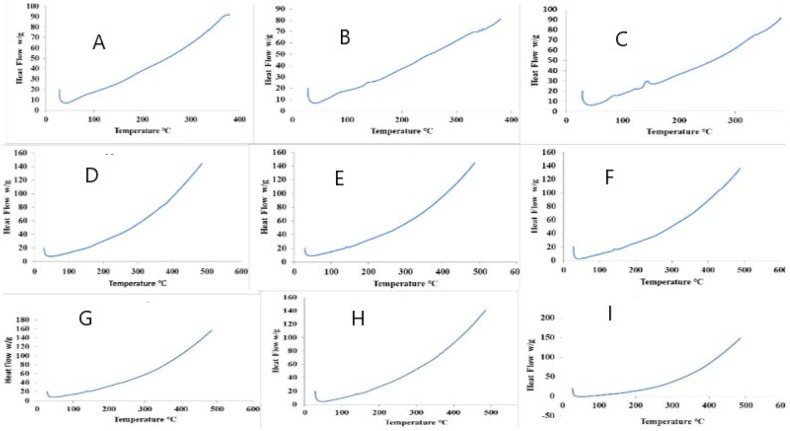
Differential scanning calorimetry graphs: (**A**) HPM35; (**B**) HPM39; (**C**) HLY45; (**D**) HLY47; (**E**) HSE55; (**F**) HSE57; (**G**) HSE59; (**H**) HSE61; (**I**) HSE58.

**Figure 9 molecules-28-03860-f009:**
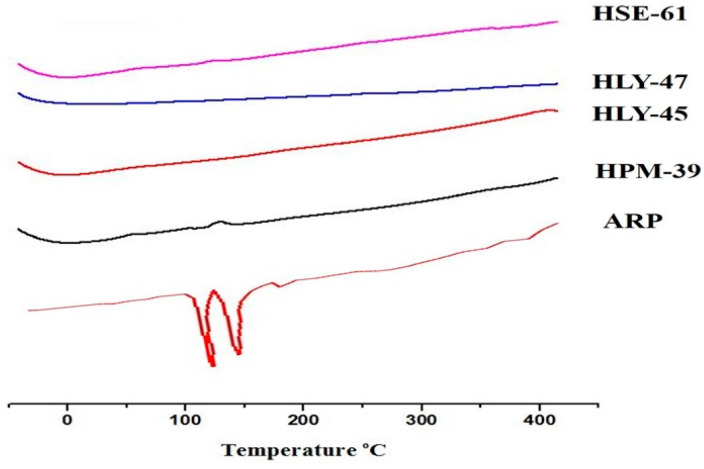
Differential scanning calorimetry of ARP, HMP39, HLY45, HLY47, and HSE61, showing HLY47 to be the best candidate for decreasing the crystalline nature of the drug.

**Table 1 molecules-28-03860-t001:** Solubility profile of ARP and binary and ternary compounds with HPβCD and LA prepared by physical mixing, solvent evaporation, and the lyophilization method. Sample HLY47 has the highest solubility.

Code	Composition	Method of Preparation	Solubility (µg/mL)	*p*-Value
ARP-1	ARP	Physical Mixing	0.04 ± 0.0	0.484235
HPM-32	ARP:HPβCD (1:1) PM	Physical Mixing	1.72 ± 0.05	0.050069
HPM-33	ARP:HPβCD (1:2.5) PM	Physical Mixing	2.99 ± 0.06	0.00349
HPM-34	ARP:HPβCD (1:4) PM	Physical Mixing	5.32 ± 0.07	1.41 × 10^−5^
HPM-35	ARP:HPβCD (1:9) PM	Physical Mixing	9.81 ± 0.02	1.39 × 10^−9^
HPM-36	ARP:HPβCD:LA (1:1:1) PM	Physical Mixing	9.88 ± 0.04	1.2 × 10^−9^
HPM-37	ARP:HPβCD:LA (1:1:0.27) PM	Physical Mixing	8.34 ± 0.03	2.1 × 10^−8^
HPM-38	ARP:HPβCD:LA (1:2.5:0.27) PM	Physical Mixing	11.012 ± 0.09	1.75 × 10^−10^
HPM-39	ARP:HPβCD:LA (1:3.6:3.6) PM	Physical Mixing	20.33 ± 0.07	1.35 × 10^−15^
HPM-40	ARP:HPβCD:LA (1:4:1) PM	Physical Mixing	13.88 ± 0.05	2.38 × 10^−12^
HPM-41	ARP:HPβCD:LA (1:9:1) PM	Physical Mixing	17.067 ± 0.02	4.35 × 10^−14^
HLY-42	ARP:HPβCD (1:1) LY	Lyophilization Method	1.12 ± 0.04	0.13768
HLY-43	ARP:HPβCD (1:2.5) LY	Lyophilization Method	3.7 ± 0.06	0.000664
HLY-44	ARP:HPβCD (1:4) LY	Lyophilization Method	11.029 ± 0.06	1.7 × 10^−10^
HLY-45	ARP:HPβCD (1:9) LY	Lyophilization Method	13.786 ± 0.05	2.71 × 10^−12^
HLY-46	ARP:HPβCD:LA (1:1:1) LY	Lyophilization Method	12.04 ± 0.05	3.4 × 10^−11^
HLY-47	ARP:HPβCD:LA (1:1:0.27) LY	Lyophilization Method	28.45 ± 0.05	1.48 × 10^−18^
HLY-48	ARP:HPβCD:LA (1:2.5:0.27) LY	Lyophilization Method	23.6 ± 0.03	6.69 × 10^−17^
HLY-49	ARP:HPβCD:LA (1:3.6:3.6) LY	Lyophilization Method	24.2 ± 0.01	4.02 × 10^−17^
HLY-50	ARP:HPβCD:LA (1:4:1) LY	Lyophilization Method	18.6 ± 0.01	7.96 × 10^−15^
HLY-51	ARP:HPβCD:LA (1:9:1) LY	Lyophilization Method	21.73 ± 0.03	3.55 × 10^−18^
HSE-52	ARP:HPβCD (1:1) SE	Solvent evaporation	1.38 ± 0.00	0.091051
HSE-53	ARP:HPβCD (1:2.5) SE	Solvent evaporation	6.6 ± 0.00	7.76 × 10^−7^
HSE-54	ARP:HPβCD (1:4) SE	Solvent evaporation	8.9 ± 0.00	7.15 × 10^−9^
HSE-55	ARP:HPβCD (1:9) SE	Solvent evaporation	10.89 ± 0.03	2.13 × 10^−10^
HSE-56	ARP:HPβCD:LA (1:1:1) SE	Solvent evaporation	20.91 ± 0.01	7.69 × 10^−16^
HSE-57	ARP:HPβCD:LA (1:1:0.27) SE	Solvent evaporation	21.63 ± 0.02	3.89 × 10^−16^
HSE-58	ARP:HPβCD:LA (1:2.5:0.27) SE	Solvent evaporation	19.42 ± 0.02	3.38 × 10^−15^
HSE-59	ARP:HPβCD:LA (1:3.6:3.6) SE	Solvent evaporation	20.9 ± 0.03	7.76 × 10^−16^
HSE-60	ARP:HPβCD:LA (1:4:1) SE	Solvent evaporation	15.1 ± 0.01	4.73 × 10^−13^
HSE-61	ARP:HPβCD:LA (1:9:1) SE	Solvent evaporation	12.63 ± 0.01	1.41 × 10^−11^

**Table 2 molecules-28-03860-t002:** Composition of binary and ternary inclusion complexes with HPβCD and LA prepared by physical mixing, solvent evaporation, and the lyophilization method.

Code	Composition
ARP-1	ARP
HPM-32	ARP:HPβCD (1:1) PM
HPM-33	ARP:HPβCD (1:2.5) PM
HPM-34	ARP:HPβCD (1:4) PM
HPM-35	ARP:HPβCD (1:9) PM
HPM-36	ARP:HPβCD:LA (1:1:1) PM
HPM-37	ARP:HPβCD:LA (1:1:0.27) PM
HPM-38	ARP:HPβCD:LA (1:2.5:0.27) PM
HPM-39	ARP:HPβCD:LA (1:3.6:3.6) PM
HPM-40	ARP:HPβCD:LA (1:4:1) PM
HPM-41	ARP:HPβCD:LA (1:9:1) PM
HLY-42	ARP:HPβCD (1:1) LY
HLY-43	ARP:HPβCD (1:2.5) LY
HLY-44	ARP:HPβCD (1:4) LY
HLY-45	ARP:HPβCD (1:9) LY
HLY-46	ARP:HPβCD:LA (1:1:1) LY
HLY-47	ARP:HPβCD:LA (1:1:0.27) LY
HLY-48	ARP:HPβCD:LA (1:2.5:0.27) LY
HLY-49	ARP:HPβCD:LA (1:3.6:3.6) LY
HLY-50	ARP:HPβCD:LA (1:4:1) LY
HLY-51	ARP:HPβCD:LA (1:9:1) LY
HSE-52	ARP:HPβCD (1:1) SE
HSE-53	ARP:HPβCD (1:2.5) SE
HSE-54	ARP:HPβCD (1:4) SE
HSE-55	ARP:HPβCD (1:9) SE
HSE-56	ARP:HPβCD:LA (1:1:1) SE
HSE-57	ARP:HPβCD:LA (1:1:0.27) SE
HSE-58	ARP:HPβCD:LA (1:2.5:0.27) SE
HSE-59	ARP:HPβCD:LA (1:3.6:3.6) SE
HSE-60	ARP:HPβCD:LA (1:4:1) SE
HSE-61	ARP:HPβCD:LA (1:9:1) SE

## Data Availability

The datasets generated during the current study are available from the corresponding author on reasonable request.

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
