# Peer review of "Enhanced Solubility and Stability of Aripiprazole in Binary and Ternary Inclusion Complexes Using Hydroxy Propyl Beta Cyclodextrin (HPβCD) and L-Arginine"

_molecules, 2023, doi:10.3390/molecules28093860_

Round 1
Reviewer 1 Report
The manuscript entitled "Enhanced solubility and stability of Aripiprazole in binary and ternary Inclusion complexes using Hydroxy Propyl beta cyclodextrin (HPβCD) and L-Arginine" has written and organized well and sounds interesting in the field, but there are some issues needs more attention.
It is recommended to describe gape of research and importance of current study in introduction.
Novelty and original features of the study have to highlight more,
Method of dissolution study and TGA/DSC analysis of samples are not clearly described.
Logic of application and discussion on Differential scanning calorimetry and TGA and AT-FTIR spectra needs more attention.
Management of results and discussion base on findings of current study through comparison to relevant studies is necessary.
More statistical analysis and finding correlations is needed.
Conclusion have to rewrite after improvement of discussion data management.
Author Response
Dear Respected Reviewer,
We have tried to address all the worthy comments. Kindly find the attached word file below.
As per reviewer 1:
Comment 1: It is recommended to describe gape of research and importance of current study in introduction.
The data in incorporated in the introduction section.
Comment 2: Novelty and original features of the study have to highlight more.
The required data is incorporated in the introduction section.
Comment 3: Method of dissolution study and TGA/DSC analysis of samples are not clearly described.
I have tried to improve the given task.
Comment 4: Logic of application and discussion on Differential scanning calorimetry and TGA and AT-FTIR spectra needs more attention.
I have tried to improve the given task.
Comment 5: More statistical analysis and finding correlations is needed.
Data has been included as suggested.
Comment 6: Management of results and discussion base on findings of current study through comparison to relevant studies is necessary.
Results and discussion
is improved.
Thanks & Best Regards
Reviewer 2 Report
The authors Sophia, et al., developed binary and ternary Inclusion complexes of Aripiprazole by solvent evaporation and Lyophilization method using hydroxypropyl-β-cyclodextrin (HP-β-CD) and L-Arginine (LA) as solubility enhancer Further the developed formulations were analyzed by a various in vitro experimental method such saturation solubility analysis, and dissolution studies. The characterization and confirmation of inclusion complexes were performed by FTIR, XRD, DSC, SEM, and TGA. The experimental results confirm that the lyophilization method is superior to solvent evaporation and also HPβCD helps to increase the solubility of the drug at a lower ratio when LA is used as an auxiliary component in a specified ratio. Although the manuscript is organized and have researcher interest. However, major caveats must be addressed before the study deserves publication.
1. Number of papers have already been reported on the improvement of Aripiprazole solubility. So, the authors should improve the introduction part with proper justification and compare the rationale of this developed Aripiprazole inclusion versus reported/published in the literature.
2. Authors should also include a few advantages or disadvantages of this inclusion complex after comparing it with other nanoformulations. Without knowing this, it is not possible to assess the novelty, and potential impact, of this paper. Also, justify how this formulation is more suitable rather than other developed formulations.
3. Authors should double-check the manufacturer name of instruments used in the method section which should be incorporated in the revised manuscript.
4. The lyophilization condition must be elaborated with reference by the authors.
5. Was the physical stability of the developed inclusion checked? Please explain.
6. Authors should include the NMR of optimized formulations. IR does not confirm the formation of the complex.
7. It would be advantageous to include in vivo Pharmacokinetics experiments in a rat model to strengthen the outcomes of the present study.
8. Authors should compare the results of table 1 by statistical program and include them in the manuscript.
9. Author should include the Mean ± SD results with an error bars in the figures.
10. Authors should improve the figure DPI quality and size in the revised manuscript.
11. Authors should elaborate more about the future perspective of this inclusion complex in the conclusion section.
12. There are many grammatical and typo errors (such as in vitro should be in italics) throughout the manuscript which should be corrected.
Author Response
Dear Respected Reviewer,
We have tried our level best to address your worthy comments. Kindly find the enclosed word file below.
- Number of papers have already been reported on the improvement of Aripiprazolesolubility. So, the authors should improve the introduction part with proper justification and compare the rationale of this developed Aripiprazole inclusion versus reported/published in the literature.
Included please
- Authors should also include a few advantages or disadvantages of this inclusion complex after comparing it with other Nano-formulations. Without knowing this, it is not possible to assess the novelty, and potential impact, of this paper. Also, justify how this formulation is more suitable rather than other developed formulations.
Included please
- Authors should double-check the manufacturer name of instruments used in the method section which should be incorporated in the revised manuscript.
Data is updated
- The lyophilization condition must be elaborated with reference by the authors.
Done please
- Was the physical stability of the developed inclusion checked? Please explain.
The samples are tested after three months by DSC, no change of graphs were seen. The developed formulations are stored in properly washed, oven dried, cold, amber colored glass bottles and placed in desiccator having silica.
- Authors should include the NMR of optimized formulations. IR does not confirm the formation of the complex.
As I belong to a developing country, it is highly requested to please accept my apologies for not able to conduct NMR studies in the specified period of 10 days. But I will include them in future studies.
- It would be advantageous to include in vivoPharmacokinetics experiments in a rat model to strengthen the outcomes of the present study.
Definitely, I will include in further studies.
- Authors should compare the results of table 1 by statistical program and include them in the manuscript.
Table 1 has the codes only. Table 2 has values and its data is compared with statistical program and included in manuscript.
- Author should include the Mean ± SD results with an error bars in the figures.
Data has been uipdated.
- Authors should improve the figure DPI quality and size in the revised manuscript.
DPI quality is improved.
- Authors should elaborate more about the future perspective of this inclusion complex in the conclusion section.
The conclusion is updated.
- There are many grammatical and typo errors (such as in vitroshould be in italics) throughout the manuscript which should be corrected.
I have tried to improve them all.
Thanks & Best Regards
Round 2
Reviewer 2 Report
Authors addressed all comments and incorporated the suggestions in the revised manuscript appropriately. I have only one minor comment.
1. Authors should remove the SD column from Table 2. It will be better to include the P value (if possible) for comparison of solubility results between final versus tested.
Author Response
Dear Respected Reviewer,
SD column has been removed from the Table 2 and the P Values have been included as per your recommendation. Please feel free to contact in case of any query.